# The study strategies of small liberal arts college students before and after COVID-19

Hailey L. Rinella, Adam L. Putnam  *

Department of Psychology, Furman University, Greenville, South Carolina, United States of America

* adam.putnam@furman.edu

## Abstract

Research has clearly demonstrated that some study strategies (for example, self-testing and spaced studying) are effective, yet students often report studying ineffectively. Our focus with the current study is to update and extend the current literature on how college students study. We surveyed 484 introductory psychology students at a small liberal arts college—a different type of school from prior studies. Our survey built on an existing study strategies questionnaire used to assess a variety of student study behaviors and beliefs. Additionally, we asked new questions about multitasking and study scheduling. Overall, we found that the current sample reported studying in similar ways to what past research suggested; students used both effective and ineffective strategies, some of which correlated with grade point average (GPA). However, some differences emerged. For example, our students were more likely to report learning how to study from a teacher. Additionally, a majority of students believed that multitasking was ineffective, yet most reported multitasking while studying. Finally, an important, but exploratory, analysis demonstrated that study strategies were similar before and after COVID-19 forced classroom changes. We highlight the need for future research on study strategies to recruit participants from more diverse institutions.

## Introduction

Effective studying is key to success in school, but doing so may be easier said than done. Most students struggle to juggle classes, extracurricular activities, and their social lives, and this juggling can lead them to study in ineffective ways, such as putting off studying to the last minute [1,2]. Although psychology research has provided clear evidence-based recommendations about how students should study, it remains unclear how often students follow these recommendations or are even aware of them [3,4]. The goal of the current study was to examine students' self-reported study behaviors, knowledge about multi-tasking, and whether students study differently as a result of the COVID-19 pandemic.

### Which study strategies are effective?

A wealth of empirical research in laboratories and classrooms over the last 20 years has clearly demonstrated that not all study strategies are equally effective (for reviews see [5–7]). On one hand, retrieval practice and spaced studying are two highly effective study strategies that help

**Data Availability Statement:** That data are posted on the Open Science Framework: OSF.IO/JMXH2.

**Competing interests:** The authors have declared that no competing interests exist.

learners of all ages and abilities to build robust, flexible, and long-lasting knowledge in many domains [5,8–10]. On the other hand, two extremely popular study strategies—highlighting and rereading—have continually been demonstrated to have small or no positive effect on learning [5,11], and may even be negatively correlated with grade point average, (GPA; [12]).

Not only have psychological scientists published this research in academic journals, but they have also broadcasted these findings to wider audiences. Tips about effective studying have been presented in books, YouTube channels, websites, and even podcasts, catered to both teachers and students [13–17]. Given both the scientific consensus identifying effective study habits and extensive outreach efforts, one might assume that students and teachers should have a good understanding of how to study, but survey research suggests otherwise.

## How do students actually study?

Research exploring what students know about effective studying has relied on two methods: (1) examining metacognitive judgments or study behavior in laboratory experiments and (2) surveying students about study habits. Both approaches suggest that students frequently make poor study choices.

Laboratory studies have consistently shown that students do not recognize the benefits of spacing and self-testing. Students typically report feeling like they learned less from using these effective strategies compared to massed study and rereading, and when given a choice, they often chose easier and less effective strategies [8,18,19]. This pattern is thought to occur largely because of fluency illusions—rereading and cramming lead to increased feelings of fluency, which learners interpret as mastery over the to-be-learned materials. In contrast, strategies requiring more upfront effort often led to higher long-term retention, a concept known as desirable difficulties [20–22]. Students apparently do not learn about the effectiveness of their study strategies from experience, likely because students interpret the effort needed for using desirable strategies as indication of their ineffectiveness [23].

Similarly, survey research indicates that students study ineffectively [3,11,19,24–28]. A recent meta-analysis revealed that rereading is the most frequently reported study strategy used (endorsed by 78% of students), and that many students also endorse using outlines, flash-cards, highlighting, and note-taking [6]. In a comprehensive review of the literature, Dunlosky and colleagues [5] rated rereading and highlighting as having low-utility, outlines and note-taking as having medium-utility, and flashcards, assuming they are used for retrieval practice, as having high-utility, suggesting that the most popular study strategies tend to produce low learning gains. Even when students use self-testing, such as with flashcards, they may not be doing so in an effort to enhance learning: several studies have demonstrated that students primarily use self-testing as a tool to gage how much they have learned [3,25,27].

One past survey study was particularly revealing, as the researchers measured study behaviors and how those behaviors correlated with GPA [3]. High achieving students (with GPAs over 3.6) reported using self-testing as a study strategy, whereas lower achieving students used self-testing to determine what they know. Furthermore, students with low GPAs typically chose to study whatever was due the soonest, instead of planning or spacing their studying, and tended to study later in the day. The results of Hartwig and Dunlosky [3] suggest that most students are not studying effectively, and that using ineffective study strategies is negatively correlated with GPA (for a replication, see [29]).

## How does multitasking affect learning?

One issue that has not yet been extensively explored in surveys of student study strategies is multitasking [30]. Multitasking has been extensively studied in the lab (often in the form of

cognitive control or task switching), and findings generally suggest that doing more than one thing at a time leads to poorer performance on all tasks (for a review see [31]). The impact of multitasking in educational contexts has increased with the proliferation of smart phones. Between 2012 and 2021, smartphone ownership increased from 39% to 85%, with 96% of the US population between the ages of 18–29 now owning a smartphone [32].

Unsurprisingly, modern students use their phones while studying, with upwards of 90% of students reporting that they multitask with media "sometimes" or "frequently" while studying (for a review, see [33,34]). Depending on the context, multitasking can either slow down studying [30], lower comprehension [30,35–38], or both. Critically, students appear to have poor metacognitive understanding of how multitasking affects their performance [39]. Furthermore, some forms of media multitasking during class (e.g., self-reported Facebook and text messaging use) have been shown to be negatively correlated to GPA and learning [30,40–42]. Despite the clear evidence for the potential negative effects of multitasking, it is unclear whether students understand how multitasking may affect their academic performance.

## Study habits before and after COVID-19

An exploratory question we addressed in the current study was whether the COVID-19 pandemic changed study behaviors. Clearly, the pandemic changed academic life in unprecedented ways [42]. Classes rapidly shifted to emergency remote delivery, and the next few months saw mixtures of fully online or hybrid formats for classes. There was variability in whether classes were synchronous or asynchronous and instructors struggled with online exam formats, trying to balance concerns about cheating with learning goals [43]. And of course, these academic changes were in addition to numerous other stressors related to the pandemic. For example, 71% of students reported an increase in stress during the pandemic [44–46]. Did the pandemic change study behaviors?

We had three competing predictions. The first is that given the tumultuous nature of moving online students might revert to using less effective, easier strategies, such as rereading, studying at the last minute, or not studying at all. Indeed, surveys indicate that students experienced a loss of motivation during the pandemic [44–47]. Additionally, they spent less time studying and more time procrastinating or embracing distractions [48], likely exacerbating student tendencies to multitask during online lectures [49]. Combined with a move towards open-note tests [43], it would be unsurprising if students endorsed less effective study strategies after the pandemic. A second prediction is that students may use more effective strategies. This might occur because students had fewer assignments and social obligations, and were able to devote more time to studying. Indeed, one survey showed that after the pandemic students were more likely to report finishing all expected readings for a class [48]. Finally, a third prediction is that study behaviors would largely remain the same before and after the pandemic. Indeed, some evidence suggests that learning outcomes and student perceptions of courses are similar between in-person and courses with online components [50].

## The current study

The goal of the current project was to examine student study behaviors and beliefs about study behaviors. We built on prior surveys [3,26,29], added questions about multitasking, and surveyed students before and after the COVID-19 pandemic. We aimed to replicate Hartwig and Dunlosky [3] for two reasons. First, their survey and many others examining study behaviors [24,26,29,51,52] surveyed students at large public institutions, raising the question of whether students at other types of schools—such as smaller private schools where student/faculty ratios are typically smaller—would have similar study behaviors and beliefs. Second, Hartwig and

Dunlosky [3] was published 10 years ago. Over the past decade, there has been a significant effort to communicate research findings about effective study strategies to students. As a result, today's students may have a better understanding of study strategies.

Overall, we predicted that our sample would report study behaviors that were consistent with past research—they would use less effective study strategies, favor massing and rereading, report inaccurate assumptions about why self-testing was important, and that self-testing would show a positive relationship with GPA. Additionally, we hypothesized that students who reported cramming before a test would be more likely to use passive study strategies, and that our sample would report multitasking while studying, despite also acknowledging that multitasking was detrimental to learning. Finally, as described above we had no strong *a priori* predictions about how the pandemic might change study behaviors.

## Method

We report how we determined our sample size, all data exclusions (if any), all manipulations, and all measures in the study. The Furman University IRB approved the study procedure.

### Participants

Four-hundred eighty-four introductory psychology students from a small liberal arts college in the Southeastern United States were recruited across four semesters between the Fall of 2019 and the Spring of 2021. Notably, the Fall of 2019 was conducted fully in-person. In the Spring of 2020, classes rapidly shifted to entirely online. The Fall of 2020 and Spring of 2021 courses were taught in a "hybrid-flex" format, where students had the option of attending classes in-person or online on a day-to-day basis. Multiple introductory psychology courses were offered each semester, and all the students in each section were invited to participate in our survey.

Our goal was to gather data from as many students as possible during our four semesters of data collection. Ages of students ranged from 18 to 22 years old ($M$ = 19.13, $SD$ = 1.09; 65% female, 34% male, <1% non-binary), and all class years were represented (45% freshman, 33% sophomores, 15% juniors, 7% seniors). The racial demographics of our sample (83% white, 7% Asian, 5% African American, 4% more than one race, and 1% other), mirrored the school's general population. For context, the school's acceptance rate is 56% and students had an average SAT score of 1309 and an ACT score of 30.

Participants received course credit for completing the survey. We excluded any participants from analyses who did not answer any questions on the survey. We did not exclude the data from any other participants, although some participants were missing a few responses to specific questions. After these exclusions, our final sample consisted of 478 students.

### Materials and procedure

Table 1 displays our survey questions, which were adapted from Hartwig and Dunlosky [3]. First, for Question 12 ("what is your GPA?") we added new response options: "I don't know", and "this is my first semester in college, so I don't have a college GPA yet" because many participants were first semester freshmen. Second, for Question 14 ("identify regularly used study strategies"), we removed the "other" option and added "use a mnemonic technique (acronyms, rhymes, memory palace, peg system)" as another answer choice. Finally, we added Questions 11 and 12 to assess multitasking opinions and behaviors.

It is important to note that we used self-reported GPA for our analyses. Although there are some systematic inaccuracies when it comes to self-reported grades, evidence suggests that there is still a reasonably strong relationship between self-reported GPA and actual GPA [53].

**Table 1.** Questionnaire response proportions for Hartwig and Dunlosky (2012), the current study, and chi-squared comparisons between the two surveys.

| Questions | Responses | H&D (2012) | Present study | Chi-squared ($\chi^2$ (df)) | Effect size ($V_C$) |
|---|---|---|---|---|---|
| 1.Would you study the way you do because a teacher (or teachers) taught you to study that way? | Yes | 36% | 52% | 20.83 (1) | .16* |
| | No | 64% | 47% | | |
| 2. How do you decide what to study next? | Whatever's due soonest/overdue | 56% | 62% | 63.40 (4) | .28* |
| | Whatever I haven't studied for the longest time | 5% | 1% | | |
| | Whatever I find interesting. | 2% | 7% | | |
| | Whatever I feel I'm doing the worst in | 24% | 8% | | |
| | I plan my study schedule ahead of time, and I study whatever I've scheduled. | 13% | 22% | | |
| 3. Do you usually return to course material to review it after a course has ended? | Yes | 23% | 23% | 0.04 (1) | .007 |
| | No | 78% | 76% | | |
| 4. All other things being equal, what do you study more for? | Essay/short answer exams | 20% | 25% | 3.36 (2) | .06 |
| | Multiple choice exams | 22% | 21% | | |
| | About the same | 58% | 53% | | |
| 5. When you study, do you typically read a textbook/article more than once? | Yes, I reread whole chapter/articles | 19% | 11% | 19.39 (2) | .16* |
| | I reread sections that I underlined highlighted or marked | 64% | 61% | | |
| | Not usually | 17% | 28% | | |
| 6. If you quiz yourself while you study (either using a quiz at the end of a chapter, or a practice quiz, or flashcards, or something else) why do you do it? | I learn more that way that I would through rereading | 27% | 30% | 15.63 (3) | .14* |
| | To figure out how well I have learned the information I'm studying | 54% | 48% | | |
| | I find quizzing more enjoyable than rereading | 10% | 17% | | |
| | I usually do not quiz myself | 9% | 4% | | |
| 7. Imagine that in the course of studying, you become convinced that you know the answer to a certain question. What would you do? | Make sure to study (or test yourself on) it again later | 46% | 43% | 0.42 (1) | .02 |
| | Put it aside and focus on other material | 54% | 56% | | |
| 8. What time of day do you most often do your studying? | Morning | <1% | 7% | 39.71 (3) | .22* |
| | Afternoon | 11% | 19% | | |
| | Evening | 69% | 49% | | |
| | Late night | 20% | 25% | | |
| 9. During what time of day do you believe your studying is (or would be) most effective? | Morning | 15% | 23% | 19.14 (3) | .15* |
| | Afternoon | 27% | 31% | | |
| | Evening | 50% | 35% | | |
| | Late night | 9% | 12% | | |
| 10. Which of the following best describes your pattern of study? | I most often space out my study sessions over multiple days/weeks | 47% | 51% | 1.72 (1) | .05 |
| | I most often do my studying in one session before the test | 53% | 48% | | |
| 11. Imagine your normal study routine. What, if any, other activities do you typically do while you're studying? *(Please check all that apply* | Listen to music | - | 60% | - | - |
| | Watch videos | - | 16% | | |
| | Communicate using your phone | - | 42% | | |
| | Browse social media | - | 26% | | |
| | Browse the web generally | - | 8% | | |
| | Talk to other people in person | - | 34% | | |
| | None of the above | - | 16% | | |

(*Continued*)

**Table 1.** (Continued)

| Questions | Responses | H&D (2012) | Present study | Chi-squared ($\chi^2$ (df)) | Effect size ($V_C$) |
|---|---|---|---|---|---|
| 12. Do you think it is more effective to focus only on what you're studying, or to multitask while you study? | It is more effective to focus only on studying | - | 90% | - | - |
| | It is more effective to multitask while studying | - | 8% | | |
| 13. What is your current grade point average? | 0.0–1.6 | 0% | 0% | - | - |
| | 1.7–2.1 | 7% | <1% | | |
| | 2.2–2.6 | 17% | 4% | | |
| | 2.7–3.1 | 24% | 14% | | |
| | 3.2–3.6 | 36% | 26% | | |
| | 3.7–4.0 | 17% | 19% | | |
| | This is my first semester in college, so I don't have a college GPA yet | - | 32% | | |
| | I don't know | - | 4% | | |
| 14. Which of the following study strategies do you use regularly? *(Please check all that apply)* | Test yourself with questions or practice problems | 71% | 75% | - | - |
| | Use flashcards | 62% | 52% | | |
| | Recopy your notes | 33% | 43% | | |
| | Reread chapters, articles, notes, lecture slides, etc | 66% | 66% | | |
| | Make outlines | 22% | 43% | | |
| | Consult online resources that were provided by the instructor | - | 51% | | |
| | Consult online resources that were NOT provided by your instructor | - | 44% | | |
| | Underline or highlight while reading | 72% | 52% | | |
| | Make diagrams, charts or pictures | 15% | 25% | | |
| | Study with friends | 50% | 51% | | |
| | Use a mnemonic technique | - | 42% | | |
| | Cram information the night before the test | 66% | 51% | | |
| | Ask questions or verbally participate during class | 37% | 37% | | |

Note.

* *p*-value < = .001, for a Chi-Squared test of independence. H & D refers to the results from Hartwig and Dunlosky [3]. Column 4 represents Cramer's *V*, a measure of effect size for categorical data that ranges from 0 to 1.0, with higher numbers representing a larger difference between the samples. S1 Table in S1 File includes the results reported above along with the results of Kornell and Bjork, (2007) and Morehead et al., (2016), both of which used a similar survey.

Overall, high achievers tend to report GPAs accurately, whereas low achievers tend to claim their GPAs are higher than they actually are [54].

All students enrolled in a section of introduction to psychology were invited to complete a large survey study. Our study skills inventory was one of several survey elements with the others addressing unrelated topics (e.g., Fear of Missing Out and disordered eating). Students saw the link to the survey posted to the subject pool website and could complete their survey from their own laptop any time before the semester ended. The entire survey took about 60 minutes.

## Results

Our preregistered analysis plan, data, and R analysis scripts are available at OSF.IO/JMXH2. Data were collected before we submitted the preregistration, but no one on the research team

had seen any of the data when the preregistration was submitted. We set alpha at .05 to determine statistical significance. Table 1 presents the main study results.

## Preregistered analyses

**What are the common behaviors and beliefs about studying among college students? How do the current results compare to prior research?.** Two of our preregistered research questions–examining the overall pattern of responses and comparing our results to prior work [3]–are addressed here. Table 1 displays the response percentages from the current survey along with the percentages reported in Hartwig and Dunlosky [3], along with the effect size of the difference and the statistical significance. Because the data were categorical, we used Chi-squared tests of independence and Cramer's $V_C$ as our measure of effect size to look for differences between the two studies. Cramer's $V_C$ ranges between 0 and 1, with higher numbers indicating a larger difference between the two sets of responses. We didn't compare responses to Question 11 and Question 12 (both about multitasking) because we added those questions to the survey, nor did we compare responses for Question 13 (GPA) and Question 14 (specific study behaviors) because we added additional response options to these questions.

As seen in Table 1 (and in S1 Table in S1 File), the current responses are generally similar to past surveys. Of the ten questions for which we conducted inferential tests, four yielded nearly identical response rates. Students from both populations ("CS"- current study, "HD"- Hartwig and Dunlosky's results) reported similar tendencies (i.e., all $p$ values were greater than .05) to: return to material after a class had ended (Question 3, HD- 23% vs. CS- 23%), studying the same amount of time for essay and short answer exams (Question 4, CS- 53% vs. HD- 58%), set aside materials they know to focus on other materials (Question 7, CS- 56% vs. HD- 54%), and intentionally spacing out their studying (Question 10, CS- 51% vs. HD- 47%).

In contrast, the other six questions revealed statistically significant differences between the current sample and Hartwig and Dunlosky [3], but in all cases the effect sizes were small, as seen in the last column in Table 1. The students in our survey were more likely to report learning how to study from a teacher (Question 1, CS- 52% vs. HD- 36%), and less likely to report studying what they were doing worst in (Question 2, CS- 8% vs. HD- 24%). Our students were less likely to report rereading whole articles (Question 5, CS- 11% vs. HD- 19%), but more likely to report not rereading at all (Question 5, CS- 28% vs. HD- 17%). Finally, our students differed in terms of when they studied and when they thought studying would be effective. Our students were less likely to study in the evening (Question 8, CS- 49% vs. HD- 69%) and more likely to study in the morning, afternoon, and late at night. Only 35% of our students thought studying in the evening would be most effective compared to 50% of students in prior research (Question 9).

Finally, although we did not calculate inferential statistics for Question 14—which questioned students about the use of specific study behaviors—we can still see in Table 1 that the overall pattern of responding is generally similar to past surveys. The notable exceptions are that our students are more likely to make outlines, less likely to underline or highlight while reading, and less likely to cram before a test.

**Why do students use self-retrieval practice as a study strategy?.** Do students know about the multiple benefits of self-testing for learning (Question 6)? Although 48% of students self-tested to keep track what they have learned, 30% reported self-testing because they learn more than they do through rereading, suggesting that these students understand the benefit of self-testing related to learning gain. These numbers were comparable to past work.

**Is the use of different study strategies related to GPA?.** We conducted two analyses to evaluate the relationships between individual study strategies and GPA. First, we computed

**Table 2. Linear regression predicting GPA and from endorsement of specific study strategies.**

| Study strategies | B | SE | CI | | P |
|---|---|---|---|---|---|
| | | | LL | UL | |
| 1. Self-test | 0.04 | 0.06 | -0.08 | 0.17 | .49 |
| 2. Use flashcards | -0.05 | 0.05 | -0.15 | 0.05 | .29 |
| 3. Recopy your notes | -0.02 | 0.05 | -0.13 | 0.08 | .69 |
| 4. Reread | -0.05 | 0.06 | -0.16 | 0.06 | .38 |
| 5.Make outlines | 0.04 | 0.05 | -0.06 | .15 | .40 |
| 8. Underline or highlight | 0.11 | 0.05 | 0.006 | .21 | .04 |
| 9. Make diagrams, charts or pictures | 0.12 | 0.07 | -0.01 | .25 | .08 |
| 10. Study with friends | -0.14 | 0.05 | -0.24 | -0.04 | .005 |
| 11. Use a mnemonic technique | 0.13 | 0.05 | 0.02 | 0.22 | .02 |
| 12. "cram" lots of information the night before the test | -0.02 | 0.05 | -0.12 | 0.08 | .72 |
| 13. Ask questions or verbally participate during class | 0.11 | 0.06 | < .001 | 0.22 | .05 |

Note. *CI* = Confidence interval, *LL* = lower limit, *UL* = upper limit.

Goodman-Kruskal gamma correlations between GPA and each of the specific study strategies listed in in Question 14 (see left column of Table 2). To do so, we coded student responses as a 0 or 1 and then correlated each strategy with self-reported GPA (we converted the response ranges to the midpoint). Critically, there were no significant correlations between any of the study strategies and GPA. These findings are inconsistent with Hartwig and Dunlosky [3], who found that self-testing was positively correlated with GPA.

Second, we ran a linear regression analysis to evaluate possible relationships between individual study strategies and GPA: doing so allowed us to simultaneously evaluate each study strategy while controlling for the Type I error rate. The regression model (see Table 2) revealed that underlining/highlighting while reading, using a mnemonic technique, and participating in class were positively associated with GPA, meaning that students who reported regularly using these study strategies had higher GPAs. In contrast, studying with friends was negatively associated with GPA, meaning that students who reported regularly studying with friends had lower GPAs. This latter finding was similar to Hartwig and Dunlosky [3]. However, Hartwig and Dunlosky also found that self-testing and rereading were positively associated with GPA and that making outlines was negatively associated with GPA, whereas we did not find a significant relationship for any of those variables.

**Does cramming before tests correlate with the use of passive study habits?.** We were interested to see if students who studied at the last minute were also likely to use less-active study techniques. Despite variability in the field regarding the definition of active and passive study strategies (for a review see [55]), we defined active study strategies as being characterized by cognitive effort and deep semantic processing (self-testing and flashcard use) whereas passive study strategies do not use such processes (highlighting, rereading, recopying notes). We calculated Goodman-Kruskal's gamma correlations between students' use of active and passive study strategies and their response to how they typically scheduled their study sessions (Question 10 –"do you space or cram?"). We found no significant correlations between the study strategies listed above and study pattern, meaning that none of the study strategies had an association with spacing or cramming.

**Do students know that multitasking while studying is ineffective? Do students report multitasking while they study?.** Our two new survey questions (Question 11 and Question 12) measured multitasking behaviors and attitudes about multitasking. Specifically, Question

11 asked students to check all common multitasking behaviors that they regularly used and Question 12 asked whether students thought multitasking harmed their studying. Although most students (90%) reported that they believed multitasking made their studying less effective, a majority of students (84%) also reported using at least one of the common multitasking behaviors. This pattern is nearly identical to that reported in prior research [38]. The most common multitasking behaviors were listening to music while studying (60%) and communicating via phone (42%). Although listening to music may or may not have detrimental effects on reading speed and memory (see [56] for a meta-analysis), phone communication most likely does [30].

## Exploratory analyses

**Did study strategies change as a result of the COVID-19 pandemic?.** Overall, we had no strong predictions concerning whether study strategies would change as a result of the COVID-19 pandemic. As outlined in the introduction, students might have more time available for studying given the restriction of social activity. Conversely, given changes in assignment types (e.g., open note tests) and additional stressors (health and financial concerns), students may resort to easier, less effective strategies.

To address this question, we compared responses from the semester before the pandemic reached the United States (Fall of 2019) to the two semesters after our institution had been fully affected by social distancing requirements (the Fall and Spring semesters during the 2020–2021 school year). We omitted responses from the Spring of 2020 because it was during this semester that our institution shifted from in-person to online classes. Table 3 Column 5 presents the results of a series of chi-squared tests of independence comparing the distribution of responses for the semesters before and after COVID-19. Notably, we found no significant difference between the study habits during the pre-COVID-19 and post-COVID-19 periods, meaning that the students in our sample reported similar study habits before and after the pandemic. However, self-reported student GPAs did moderately rise between the pre- and post-pandemic semesters ($V = .20$). One explanation for this finding is that instructors adjusted assignments as they transitioned courses to online format (for example, students typically score higher on open-note online exams compared to traditional close-book in person exams; [43]).

**Does multitasking behavior predict whether students cram or space their studying?.** We hypothesized that students who reported using none of the multitasking behaviors listed in Question 12 would be more likely to space their studying, because these students may have more knowledge of effective study habits. Likewise, we predicted that students who reported multitasking—for example, using the phone or talking to people while studying—would be more likely to cram. We ran a binary multiple logistic regression with study pattern (Question 10, "do students study the night before or space out studying") as the outcome variable and multitasking activities (Question 12, check all multitasking activities you do regularly) as the predictor variables. The multitasking activities were either 0 (I do this activity) or 1 (I do not do this activity).

Table 4 shows the results of our regression model. Three predictors—listening to music, talking with people, and answering none of the above—predicted cramming tendencies, meaning that students who reported doing these behaviors were more likely to cram. Examining the odds ratios suggested that students who listened to music while studying were 1.77 times more likely to cram, students who talk to people while studying are 1.70 times more likely to cram, and students who reported not multitasking were 2.65 times more likely to cram. In contrast, surfing the web was the sole significant predictor of spacing, such that

**Table 3. Response percentages pre and post COVID-19.**

| Questions | Responses | Pre-COVID | Post-COVID | Chi-squared ($\chi^2$ (df)) | Effect size ($V_C$) |
|---|---|---|---|---|---|
| 1.Would you study the way you do because a teacher (or teachers) taught you to study that way? | Yes | 56% | 50% | 1.39 (2) | .06 |
| | No | 43% | 49% | | |
| 2. How do you decide what to study next? | Whatever's due soonest/overdue | 62% | 62% | 3.79 (4) | .10 |
| | Whatever I haven't studied for the longest time | 2% | 1% | | |
| | Whatever I find interesting. | 9% | 6% | | |
| | Whatever I feel I'm doing the worst in | 10% | 8% | | |
| | I plan my study schedule ahead of time, and I study whatever I've scheduled. | 18% | 24% | | |
| 3. Do you usually return to course material to review it after a course has ended? | Yes | 24% | 23% | < .001 (1) | < .001 |
| | No | 76% | 76% | | |
| 4. All other things being equal, what do you study more for? | Essay/short answer exams | 21% | 29% | 4.21 (2) | .11 |
| | Multiple choice exams | 18% | 21% | | |
| | About the same | 61% | 50% | | |
| 5. When you study, do you typically read a textbook/article more than once? | Yes, I reread whole chapter/articles | 11% | 10% | 3.01 (4) | .08 |
| | I reread sections that I underlined highlighted or marked | 58% | 64% | | |
| | Not usually | 31% | 26% | | |
| 6. If you quiz yourself while you study (either using a quiz at the end of a chapter, or a practice quiz, or flashcards, or something else) why do you do it? | I learn more that way that I would through rereading | 37% | 29% | 2.39 (3) | .08 |
| | To figure out how well I have learned the information I'm studying | 44% | 50% | | |
| | I find quizzing more enjoyable than rereading | 15% | 17% | | |
| | I usually do not quiz myself | 4% | 5% | | |
| 7. Imagine that in the course of studying, you become convinced that you know the answer to a certain question. What would you do? | Make sure to study (or test yourself on) it again later | 44% | 43% | 0 (1) | 0 |
| | Put it aside and focus on other material | 56% | 56% | | |
| 8. What time of day do you most often do your studying? | Morning | 4% | 8% | 3.47 (3) | .10 |
| | Afternoon | 18% | 20% | | |
| | Evening | 51% | 49% | | |
| | Late night | 27% | 23% | | |
| 9. During what time of day do you believe your studying is (or would be) most effective? | Morning | 21% | 24% | 4.16 (3) | .11 |
| | Afternoon | 30% | 32% | | |
| | Evening | 41% | 31% | | |
| | Late night | 8% | 12% | | |
| 10. Which of the following best describes your pattern of study? | I most often space out my study sessions over multiple days/weeks | 56% | 50% | 0.70 (1) | .05 |
| | I most often do my studying in one session before the test | 44% | 49% | | |
| 11. Imagine your normal study routine. What, if any, other activities do you typically do while you're studying? *(Please check all that apply* | Listen to music | 63% | 58% | 10.35 (13) | .07 |
| | Watch videos | 22% | 14% | | |
| | Communicate using your phone | 43% | 42% | | |
| | Browse social media | 27% | 26% | | |
| | Browse the web generally | 11% | 9% | | |
| | Talk to other people in person | 46% | 33% | | |
| | None of the above | 15% | 17% | | |

*(Continued)*

**Table 3.** (Continued)

| Questions | Responses | Pre-COVID | Post-COVID | Chi-squared ($\chi^2$ (df)) | Effect size ($V_C$) |
|---|---|---|---|---|---|
| 12. Do you think it is more effective to focus only on what you're studying, or to multitask while you study? | It is more effective to focus only on studying | 93% | 91% | 0.03 (1) | .01 |
| | It is more effective to multitask while studying | 7% | 8% | | |
| 13. What is your current grade point average? | 0.0–1.6 | 0% | 0% | 13.69 (5) | .20* |
| | 1.7–2.1 | 2% | 0% | | |
| | 2.2–2.6 | 3% | 5% | | |
| | 2.7–3.1 | 15% | 17% | | |
| | 3.2–3.6 | 29% | 26% | | |
| | 3.7–4.0 | 11% | 25% | | |
| | This is my first semester in college, so I don't have a college GPA yet | 37% | 25% | | |
| | I don't know | 3% | 3% | | |
| 14. Which of the following study strategies do you use regularly? *(Please check all that apply)* | Test yourself with questions or practice problems | 73% | 77% | 18.94 (25) | .06 |
| | Use flashcards | 60% | 50% | | |
| | Recopy your notes | 44% | 42% | | |
| | Reread chapters, articles, notes, lecture slides, etc. | 57% | 69% | | |
| | Make outlines | 47% | 43% | | |
| | Consult online resources that were provided by the instructor | 48% | 53% | | |
| | Consult online resources that were NOT provided by your instructor | 43% | 46% | | |
| | Underline or highlight while reading | 49% | 55% | | |
| | Make diagrams, charts or pictures | 30% | 22% | | |
| | Study with friends | 58% | 50% | | |
| | Use a mnemonic technique | 50% | 40% | | |
| | Cram information the night before the test | 50% | 51% | | |
| | Ask questions or verbally participate during class | 40% | 38% | | |

Note.

* *p*-value < = .001, for a Chi-Squared test of independence. Pre-COVID column includes the Fall semester of 2019 and the post-COVID column includes Fall of 2020 and the Spring of 2021.

students that reported surfing the internet were 2.30 times more likely to space their studying. Thus, in contrast to our hypothesis, one of the strongest predictors of cramming was not multitasking at all. One explanation for this pattern is that when students are cramming, they are pressed for time and thus are less likely to indulge in multi-tasking behavior.

## Discussion

The goal of the current study was to measure students' study behaviors and their beliefs about studying at a small liberal arts college. Overall, our results replicated past research (e.g., Hartwig & Dunlosky, [3]) with a few key exceptions (described below). These differences might have occurred because the current sample is different than past research, or because outreach efforts by cognitive psychologists (e.g. in the forms of books [14], podcasts [16], and blogs

Table 4. Logistic regression analysis predicting study patterns with multitasking activities.

| Variable | B (SE) | 95% CI for odds ratio | | |
|---|---|---|---|---|
| | | Lower | Odds ratio | Upper |
| 1. Listen to music | - 0.57 (0.25)* | 1.10 | 1.77 | 2.88 |
| 2. Watch videos | - 0.19 (0.28) | 0.69 | 1.21 | 2.10 |
| 3. Phone Communication | - 0.21 (0.22) | 0.80 | 1.23 | 1.91 |
| 4. Browse social media | - 0.45 (0.25) | 0.97 | 1.56 | 2.54 |
| 5. Browse the web | 0.84 (0.38)* | 0.20 | 0.43 | 0.89 |
| 6. Talk to people | - 0.53 (0.22)* | 1.11 | 1.70 | 2.63 |
| 7. None of the above | - 0.97 (0.35) ** | 1.35 | 2.65 | 5.26 |

Note.

* $p < 0.05$

** p < .001, SE = standard error, Positive B values indicate strategy use predicts spaced studying and negative B values indicate strategy use predicts cramming.

[17]) have been effective in changing study habits. Furthermore, the current study also demonstrated that most students recognize the problems with multitasking while studying, yet continue to do so. Finally, it also demonstrated that study habits before and after the onset of the COVID-19 pandemic did not differ.

## How do the current results compare to past results?

In general, our results were consistent with past work [3,26,29]. Taken together, these studies paint a picture of students who use a mix of empirically effective and less effective strategies. Many students test themselves with practice problems and use flashcards, and about half of students from all studies report spacing their studying, rather than exclusively cramming (see S1 Table in S1 File). At the same time, students study whatever is due soonest instead of scheduling their studying, rarely return to course material after a class has ended, frequently reread course materials, and use quizzing primarily to determine what they know, instead of using it to aid learning.

There were some differences between the current study and prior work. First, 52% of our students reported learning how to study from a teacher, compared to 36% (reported percentage in both [3] and [29]). Second, our sample was less likely to choose to study what they felt they were doing the worst in, were less likely to reread class materials, were more likely to find self-testing enjoyable, were more likely to study throughout the day instead of studying just in the evening, and to think that studying can be effective at all times of the day. Additionally, in the current study (and in [29]), self-testing or using flashcards was not related to GPA, whereas there was a positive correlation in [3].

What might explain these differences? One possibility is that the surge in evidence-based study advice that began in the mid-2010s could have successfully proliferated to students and teachers, such that the students in the current sample may have been exposed to more empirically-supported study advice. The increased proportions of students who reported learning study techniques from a teacher supports this explanation. Alternatively, prior studies have drawn students from large public institutions (with over 20,000 students enrolled) compared to the current study where we drew students from a small private liberal arts college (about 2,400 students enrolled). Differences in the institutions (the student to faculty ratio, program offerings, proportion of classes taught by tenured professors, and student characteristics such as high school experience) may have contributed to the differences in study habits. Future research should draw participants from a wide diversity of institutions before generalizing conclusions to students from all types of institutions.

## What do students understand about multitasking?

Similar to Pashler and colleagues' [38] findings, our survey reported that 84% of students multitask while studying despite 90% of students recognizing that their studying would be more efficient without it. This discrepancy between what students believe should be effective and what they actually do suggests further education or outreach about how to study might not be helpful. Rather, students may benefit more from advice that suggests concrete ways to *implement* effective study strategies [4].

Furthermore, an exploratory analysis suggested that students who reported not multitasking at all were more likely to cram (likely because they were under intense time pressure) and that talking to people and listening to music were also associated with cramming. In contrast, students who surfed the web, were more likely to space, perhaps because they are not feeling the same time pressure as students who report cramming. Given the clear laboratory evidence about the perils of multitasking (e.g., [38]) and the exploratory nature of these analyses, future research should explore more directly how different study behaviors are associated with multitasking study scheduling.

## COVID-19

Finally, a key finding from our paper was that student study strategies seemed to be unaffected by COVID-19: across all of our questions student responses were regardless of whether data was collected before or after social distancing restrictions were implemented. This outcome is inconsistent with survey research suggesting that students experienced a lack of motivation as a result of the pandemic [48]. Why did this discrepancy occur? One possibility is that our comparisons were looking at semesters before the pandemic and after the Spring 2020 semester when courses shifted rapidly online. Thus, the later semesters were a combination of online and hybrid formats; regardless, instructors likely had more time to prepare for teaching in hybrid formats, as opposed to the remote emergency instruction that occurred in the Spring of 2020.

## Limitations and future directions

There are a few limitations that constrain the conclusions we can draw from this study. By design, we used broad survey questions. These could have been misinterpreted. For example, students may report every study strategy they have ever used rather than the ones they use most frequently. Likewise, we asked about "multitasking" generally, but, as shown in our analyses, some types of multitasking appear to predict study scheduling behavior, whereas others do not. Future studies should focus on *specific* study strategies, rather than broad behaviors. More specific survey questions can allow for more precise inferences, and ensure that researchers and students are interpreting terms in the same way (see for example, [52]).

A second limitation is our sample composition. Our sample was drawn from a single small liberal arts college. Because some differences between the current work and prior work emerged, it is critical for future researchers to gather data from a larger range of institutions to examine how and why study behaviors might differ across the many kinds of educational institutions. As well, recruiting participants from more diverse institutions will allow for a more nuanced analysis of variations in student characteristics (e.g., full-time versus part-time) and how those variations correlate with study strategy choice. Our sample also consisted of only students from introduction to psychology courses, rather than a random sample of our entire study body. Although this is a clear limitation in some ways, by our estimates only about 25% of the students in our sample went on to become psychology majors; thus, the remaining 75% of students were taking introduction to psychology as a general education requirement or as

an elective. Regardless, further studies should recruit a variety of college majors in addition to recruiting for various general student characteristics, like course load and personal obligations. This can allow for more generalized findings, and even begin to explore whether different study strategies are more effective for different majors.

## Conclusion

The current study supports the idea that, in general, students have room to improve with their study strategies. Teachers can be a great resource and facilitator for reshaping students' incorrect preconceived notions about what is effective, but some research suggests that teachers do not know much more about effective studying than the average student [29]. Only 52% of Furman students reported studying using methods that were taught by a teacher, but this is a higher percentage than past research (specifically Hartwig and Dunlosky [3] reported that only 36% of their sample were taught study strategies by a teacher). Thus, both teachers and students can benefit from improved understanding of empirically supported study strategies.

Other scholars have explored what methods can be used within an educational setting to promote student success; critically, such programs emphasize not just what strategies are effective, but provide concrete recommendations about how to use those strategies. For example, a college course was created and tested by McDaniel and colleagues [4] for teaching the knowledge, belief, commitment, and planning (KBCP) framework (which integrates metacognition and empirical research) to teach college students to understand the mechanics behind each study strategy and how to use these strategies for effective studying. They found that the course was successful and prompted the long-term use of effective behaviors in the students' future courses, suggesting that students may benefit from receiving interventions that also address and encourage the best ways to implement changes in study behavior.

In sum, our results are consistent with prior research in showing that while students do engage in some empirically-supported study techniques, there are also many ways in which they fall short. As one example, many students report multitasking while studying, despite recognizing that such behavior is detrimental to their overall success. Studying in college is never easy, but the current results suggest that students do have room for improvement of terms of how and when they study.

## Supporting information

**S1 File. Additional results for the current study and comparison to prior studies.** S1 Table compares the current results to three prior examinations of student study strategies. S2 Table provides a detailed analysis of student study strategies before and after COVID-19. (DOCX)

## Acknowledgments

We thank Will Deng, Sam Gary, and Jason Hayden for their help on this project.

## Author Contributions

**Conceptualization:** Hailey L. Rinella.

**Data curation:** Adam L. Putnam.

**Formal analysis:** Hailey L. Rinella, Adam L. Putnam.

**Methodology:** Adam L. Putnam.

**Project administration:** Adam L. Putnam.

**Software:** Adam L. Putnam.

**Supervision:** Adam L. Putnam.

**Writing – original draft:** Hailey L. Rinella.

**Writing – review & editing:** Adam L. Putnam.

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
