## [Decision Letter · Decision Letter 0]

14 Sep 2022

PONE-D-22-20373The study strategies of small liberal arts college students before and after COVID-19PLOS ONE

Dear Dr. Putnam,

Thank you for submitting your manuscript to PLOS ONE. After careful consideration, we feel that it has merit but does not fully meet PLOS ONE’s publication criteria as it currently stands. Therefore, we invite you to submit a revised version of the manuscript that addresses the points raised during the review process.

We look forward to receiving your revised manuscript.

Kind regards,

Gwo-Jen Hwang

Academic Editor

PLOS ONE

Journal Requirements:

Reviewers' comments:

Reviewer's Responses to Questions

**Comments to the Author**

1. Is the manuscript technically sound, and do the data support the conclusions?

Reviewer #1: Yes

Reviewer #2: Yes

2. Has the statistical analysis been performed appropriately and rigorously? 

Reviewer #1: Yes

Reviewer #2: Yes

3. Have the authors made all data underlying the findings in their manuscript fully available?

Reviewer #1: Yes

Reviewer #2: Yes

4. Is the manuscript presented in an intelligible fashion and written in standard English?

Reviewer #1: Yes

Reviewer #2: Yes

5. Review Comments to the Author

Reviewer #1: In this paper, the authors examined the self-reported study behaviours, knowledge about multitasking, and study strategies used before and during COVID of 484 introductory psychology students at a small liberal arts college in the Southeastern US recruited over four semesters between 2019 and 2021.

The abstract ought to include the sample size and some of its characteristics.

Where there is statistical significance, inferential statistics (Chi-square… and effect size) should be reported. If “more or less likely” refer to odds ratios, then odd ratios should also be reported. The authors are inconsistent in that these statistics are reported in one part but withheld in other parts of the analysis.

The statement in the first paragraph of the Discussion, “because outreach efforts by cognitive psychologists..” is ambiguous and needs further explanation.

I suggest that the entire sample from an introductory psychology class only is also a limitation

It would have been helpful to suggest a few reasons that may explain why students insist on using ineffective study strategies. While I appreciate that these variables were not measured, some speculation as to their role may be helpful. For example, would there be a difference in study strategies between those who take five courses and those who take fewer than three courses, those who work full-time and those who work part-time or do not work at all, or those with family responsibility and those without such obligation?

Overall, I found the manuscript informative, useful, and easy to read and follow. I am planning to share it with my students once published. Thank you for giving me this opportunity to review this work.

Reviewer #2: Some modification to the abstract need to be done.

adding short introduction that include the focus of the current study. what the main objects of the study.

a brief description of the methods, for example participants and questionnaires

6. PLOS authors have the option to publish the peer review history of their article (what does this mean?). If published, this will include your full peer review and any attached files.

Reviewer #1: **Yes: **Dr. Ali M. AL-Asadi

Reviewer #2: No

---

## [Author Response · Author response to Decision Letter 0]

28 Sep 2022

Adam L. Putnam

Assistant Professor

Department of Psychology

Furman University

Greenville SC 29613

adam.putnam@furman.edu

864.294.3213

September 28, 2022 

Dear Gwo-Jen Hwang,

Thank you for forwarding the reviews of our manuscript, The study strategies of small liberal arts college students before and after COVID-19. We’ve included the reviewer responses below, along with our responses to their suggestions in bold.

Reviewer #1

In this paper, the authors examined the self-reported study behaviours, knowledge about multitasking, and study strategies used before and during COVID of 484 introductory psychology students at a small liberal arts college in the Southeastern US recruited over four semesters between 2019 and 2021.

The abstract ought to include the sample size and some of its characteristics.

We now report the sample size and its main composition (introductory psychology students) in the abstract.

See also the comments from Reviewer #2 and our response

Where there is statistical significance, inferential statistics (Chi-square… and effect size) should be reported. If “more or less likely” refer to odds ratios, then odd ratios should also be reported. The authors are inconsistent in that these statistics are reported in one part but withheld in other parts of the analysis.

Every time we report a comparison between our work and prior work the results of the statistical tests are now reported in Table 1. Although we often use the language “our students were more/less likely to XYZ than prior samples” we did not conduct a logistic regression analysis for those comparisons, so there is no odds ratio to report.

In the one results section where we do report odds ratios (Does multitasking behavior predict whether students cram or space their studying?) we conducted a logistic regression, thus reporting the odds ratio is appropriate. For clarification, we also added the statistical results reported in Table 4.

The statement in the first paragraph of the Discussion, “because outreach efforts by cognitive psychologists..” is ambiguous and needs further explanation.

We reiterate the point made in the introduction by re-citing relevant works about effective studying that are geared towards the general public and students. 

I suggest that the entire sample from an introductory psychology class only is also a limitation

We added the fact that our sample came from introductory psychology classes as an additional limitation.

It would have been helpful to suggest a few reasons that may explain why students insist on using ineffective study strategies. While I appreciate that these variables were not measured, some speculation as to their role may be helpful. For example, would there be a difference in study strategies between those who take five courses and those who take fewer than three courses, those who work full-time and those who work part-time or do not work at all, or those with family responsibility and those without such obligation?

In the introduction to the manuscript, we review the literature that has examined why students use ineffective study strategies. As we note, the main hypothesis in the field is that ineffective strategies – cramming, rereading, and highlighting – typically lead to increased feelings of fluency which lead students to think they have mastered the material. In contrast, effective strategies – spacing and testing -- are desirable difficulties in that they are objectively more challenging to complete initially, yet led to long term durable learning. This disconnect between a student’s experience of a technique and its overall effectiveness are likely the main reason students do not use more effective strategies.

Reviewer 1 also raises the interesting question of how different types of students might approach studying in different ways. Our institution is a small liberal arts college, and our student body is over 95% residential, full-time students of traditional college age. In short, we are not well equipped to speculate about that specific suggestion. However, we now note in the discussion that such research questions are a prime area for future research.

Overall, I found the manuscript informative, useful, and easy to read and follow. I am planning to share it with my students once published. Thank you for giving me this opportunity to review this work.

Thank you for taking the time to review our manuscript – we are glad it will be useful for you!

Reviewer #2

Some modification to the abstract need to be done.

adding short introduction that include the focus of the current study. what the main objects of the study.

a brief description of the methods, for example participants and questionnaires

We revised the abstract to more clearly state the focus of the current study, the main objectives of the study, and now include a brief description of the method (more information about the participants and the survey instrument). We refrained from citing the exact survey used in our study to follow the PLOS One submission guidelines from including citations in abstracts.

In addition to the requested changes made above, we also made several small stylistic changes (or corrected typos) that we will not bother to report here.

We hope this revised version of our manuscript meets the publication criteria for PLOS One.

Adam L. Putnam, PhD 

Assistant Professor of Psychology 

Furman University

---

## [Decision Letter · Decision Letter 1]

22 Nov 2022

The study strategies of small liberal arts college students before and after COVID-19

PONE-D-22-20373R1

Dear Dr. Putnam,

We’re pleased to inform you that your manuscript has been judged scientifically suitable for publication and will be formally accepted for publication once it meets all outstanding technical requirements.

Kind regards,

Simone Battaglia

Guest Editor

PLOS ONE

Additional Editor Comments (optional):

Reviewers' comments:

Reviewer's Responses to Questions

**Comments to the Author**

1. If the authors have adequately addressed your comments raised in a previous round of review and you feel that this manuscript is now acceptable for publication, you may indicate that here to bypass the “Comments to the Author” section, enter your conflict of interest statement in the “Confidential to Editor” section, and submit your "Accept" recommendation.

Reviewer #1: All comments have been addressed

Reviewer #2: All comments have been addressed

2. Is the manuscript technically sound, and do the data support the conclusions?

Reviewer #1: Yes

Reviewer #2: Yes

3. Has the statistical analysis been performed appropriately and rigorously? 

Reviewer #1: Yes

Reviewer #2: Yes

4. Have the authors made all data underlying the findings in their manuscript fully available?

Reviewer #1: Yes

Reviewer #2: Yes

5. Is the manuscript presented in an intelligible fashion and written in standard English?

Reviewer #1: Yes

Reviewer #2: Yes

6. Review Comments to the Author

Reviewer #1: This area of research needs much more attention than it has been given. Thank you for responding to and incorporating my comments into your manuscript.

Reviewer #2: (No Response)

7. PLOS authors have the option to publish the peer review history of their article (what does this mean?). If published, this will include your full peer review and any attached files.

Reviewer #1: **Yes: **Dr. Ali M. AL-Asadi

Reviewer #2: No

---

## [Editor Report · Acceptance letter]

29 Nov 2022

PONE-D-22-20373R1 

The study strategies of small liberal arts college students before and after COVID-19 

Dear Dr. Putnam:

I'm pleased to inform you that your manuscript has been deemed suitable for publication in PLOS ONE. Congratulations! Your manuscript is now with our production department. 

Kind regards, 

on behalf of

Dr. Simone Battaglia 

Guest Editor

PLOS ONE